

# Assessing Arctic low-level clouds and precipitation from above - a radar perspective

Imke Schirmacher[1], Pavlos Kollias[2,3], Katia Lamer[2], Mario Mech[1], Lukas Pfitzenmaier[1], Manfred Wendisch[4], and Susanne Crewell[1]

[1]Institute for Geophysics and Meteorology, University of Cologne, Germany
[2]Department of Environmental and Climate Sciences, Brookhaven National Laboratory, Stony Brook, NY, USA
[3]School of Marine and Atmospheric Sciences, Stony Brook University, Stony Brook, NY, USA
[4]Institute for Meteorology, Leipzig University, Germany

**Correspondence:** Imke Schirmacher (Imke.Schirmacher@uni-koeln.de)

**Abstract.** Most Arctic clouds occur below 2 km altitude as revealed by CloudSat satellite observations. However, recent studies suggest that the relatively coarse spatial resolution, low sensitivity, and blind zone of the radar installed on CloudSat may not enable it to comprehensively document low-level clouds. We investigate the impact of these limitations on the Arctic low-level cloud fraction, which is the amount of cloudy points with respect to all points as a function of height, derived from CloudSat radar observations. For this purpose, we leverage highly resolved vertical profiles of low-level cloud fraction derived from downlooking Microwave Radar/radiometer for Arctic Clouds (MiRAC) radar reflectivity measurements. MiRAC has been operated during four aircraft campaigns taking place in the vicinity of Svalbard during different times of the year and covering more than 25,000 km. This allows us to study the dependence of CloudSat limitations on different synoptic and surface conditions.

A forward simulator converts MiRAC measurements to synthetic CloudSat radar reflectivities. These forward simulations are compared with the original CloudSat observations for four satellite underflights to prove the suitability of our forward-simulation approach. Above CloudSat's blind zone of 1 km and below 2.5 km, the forward simulations reveal that CloudSat would overestimate the MiRAC cloud fraction over all campaigns by about 6 percent points (pp) due to its horizontal resolution, by 12 pp due to its range resolution, and underestimate it by 10 pp due to its sensitivity. Especially during cold air outbreaks over open water, high reflectivity clouds appear below 1.5 km, which are stretched by CloudSat's pulse length causing the forward-simulated cloud fraction to be 16 pp higher than that observed by MiRAC. The pulse length merges multilayer clouds, whereas thin low-reflectivity clouds remain undetected. Consequently, 48 % of clouds observed by MiRAC belong to multilayer clouds, which reduces by a factor of 4 for the forward-simulated CloudSat counterpart. Despite the overestimation between 1 and 2.5 km, the overall low-level cloud fraction is strongly reduced due to CloudSat's blind zone that misses a cloud fraction of 32 % and half of the total (mainly light) precipitation amount.





## 1 Introduction

Low-level clouds are prominent features of the Arctic climate (Shupe et al., 2006; Liu et al., 2012; Mioche et al., 2015) that have a large impact on the radiative energy budget of the Arctic surface (e.g., Curry et al., 1996; Shupe and Intrieri, 2004;

Wendisch et al., 2019). In contrast to the global cooling effect of low clouds, between 70–82° N they may create a positive (warming) cloud radiative forcing (CRF) of $\sim 10\,\mathrm{W\,m^{-2}}$ (Kay and L'Ecuyer, 2013). The terrestrial CRF dominates and warms the near-surface air due to low solar elevation during polar day and absent solar radiation during polar night (Lubin and Vogelmann, 2006; Stapf et al., 2021). Within the last four decades, the near-surface warming in the Arctic increased stronger than on global average called Arctic amplification (Serreze and Barry, 2011; Wendisch et al., 2019). Diverse processes and

interacting feedback mechanisms lead to Arctic amplification. An increased cloud cover and amount of water vapor and the lapse rate feedback from persistent clouds (Graversen et al., 2008) would enhance the terrestrial downward radiation (Francis and Hunter, 2006) and contribute stronger to Arctic amplification than the sea ice-albedo feedback (Winton, 2006). Thus, there is high interest on accurate observations of Arctic low-level cloud properties and their changes.

Detailed ground-based remote sensing observations that measure the vertical distribution and variability of low-level clouds

are available from very few stations in the Arctic (e.g., Liu et al., 2017; Gierens et al., 2020). They allow us to study the temporal variability over long time periods but at a specific place. In contrast, ship campaigns into the high Arctic (Shupe et al., 2022; Intrieri and Shupe, 2004) assess the spatial variability only over short time periods but over a larger yet still limited area. As recently highlighted by Griesche et al. (2021) using measurements from the RV *Polarstern* in the Marginal sea Ice Zone (MIZ), however, the frequent occurrence of low-level stratus around 100 m is often missed by ground-based observations. On larger

spatial scales, active satellite measurements resolve vertical cloud structures for long time periods. CloudSat (Stephens et al., 2002) has been frequently used for studies of Arctic clouds such as for investigating the correlation between low-level cloud occurrence and sea ice concentration (Zygmuntowska et al., 2012; Mioche et al., 2015). For the years 2006 to 2011, Liu et al. (2012) find that roughly 80 % of clouds over the Arctic Ocean occur below 2 km altitude.

CloudSat and ground-based observations at the Eureka site, Canada, revealed different cloud occurrences below 2 km altitude

(Blanchard et al., 2014; Mioche et al., 2015). Thus, it remains to be determined if CloudSat captures all low-level Arctic clouds due to its limitations: First, CloudSat's along-track sampling conceals spatial cloud patterns. Second, according to Lamer et al. (2020), the pulse length either stretches or fails to detect shallow clouds. Third, the lowest levels from CloudSat's vertical profiles suffer from ground clutter due to reflections at the surface called blind zone. This blind zone prevents the cloud assessment roughly below the first kilometer (Palerme et al., 2019; Lamer et al., 2020; Liu, 2022). Using ground-based radar

measurements as reference, Maahn et al. (2014) showed that CloudSat underestimates the total precipitation by 9 percent points (pp) over Ny-Ålesund, Svalbard. The representation of Arctic low-level clouds in climate models is of high relevance to investigate for example its correlation with sea ice concentration (Morrison et al., 2019). To fully exploit CloudSat for improving climate models it is necessary to know its limitations and thus to evaluate CloudSat measurements with more fine resolved observations that ideally cover broad areas over land and ocean. These measurements should ultimately address how



low-level cloud occurrence varies close to surface, depends on surface characteristics and meteorological situation, and thus affects Arctic amplification.

CloudSat observations have been compared with airborne remote sensing (e.g., Gayet et al., 2009; Painemal et al., 2019) for relatively homogeneous clouds at higher altitudes to calibrate airborne instruments (Barker et al., 2008; Protat et al., 2009, 2011). For the first time, Liu (2022) investigates synthetic CloudSat cloud masks in the Arctic region. These data are

based on radar reflectivities from QuickBeam radar forward simulations (Haynes et al., 2007) that used vertical profiles of retrieved cloud properties from ground-based radar and lidar during the SHEBA (Surface Heat Budget of the Arctic Ocean) experiment. Compared to ground-based observations, the forward-simulated data detected all clouds with height above 1 km, but 25 pp less below 600 m. Nevertheless, in this study the synthetic data were generated under several assumptions and by low-temporal resolution measurements that had a different viewing geometry.

In this study we investigate vertical profiles of low-level cloud occurrences over the Fram Strait using CloudSat observations and measurements by the airborne Microwave Radar/radiometer for Arctic Clouds (MiRAC; Mech et al., 2019) operating at the same radar wavelength as CloudSat. MiRAC measured highly resolved profiles with a lower blind zone of about 150 m onboard the Polar 5 (Wesche et al., 2016) research aircraft during four airborne campaigns, that have been conducted in the vicinity of Svalbard within the framework of the German DFG project - TRR 172, "ArctiC Amplification: Climate Relevant Atmospheric

and SurfaCe Processes, and Feedback Mechanisms" ((AC)[3]; Wendisch et al., 2023). The Svalbard region is of particular interest because the steady heat and moisture flux of the North Atlantic Ocean enhances cloud fraction and precipitation compared to the entire Arctic (Mioche et al., 2015; McCrystall et al., 2021). The campaigns, namely ACLOUD (Arctic CLoud Observations Using airborne measurements during polar Day; Wendisch et al., 2019; Ehrlich et al., 2019), AFLUX (Airborne measurements of radiative and turbulent FLUXes of energy and momentum in the Arctic boundary layer; Mech et al., 2022), MOSAiC-ACA

(Multidisciplinary drifting Observatory for the Study of Arctic Climate-Airborne observations in the Central Arctic; Mech et al., 2022), and HALO-(AC)[3] (High Altitude and LOng range research aircraft - (AC)[3]) covered periods from 2017 to 2022 between March and September. Since Polar 5 flies relatively slow, a unique database has been gathered that covers more than 25,000 km and includes four underflights of Polar 5 below CloudSat. The larger spatial coverage of the airborne observations compared to observations from land stations allows for new insights into the cloud variability over open ocean and sea ice.

The manuscript is organized as follows: First, we present the CloudSat and airborne remote sensing data and describe how CloudSat's radar reflectivities are forward simulated from MiRAC observations (Sect. 2). Second, we outline the meteorological situation encountered during the campaigns (Sect. 3). Section 4 evaluates the forward simulations for four underflights and investigates the effects of CloudSat's spatial resolution, blind zone, and sensitivity on its performance to detect low-level clouds. Afterwards, Sect. 5 compares the fraction of the MiRAC and forward-simulated radar reflectivities across the entire

data with height to analyze the variability of low-level Arctic cloud occurrence with respect to meteorological and surface conditions and to identify states that limit CloudSat's cloud detection the most. Section 6 concludes the study and discusses future steps.





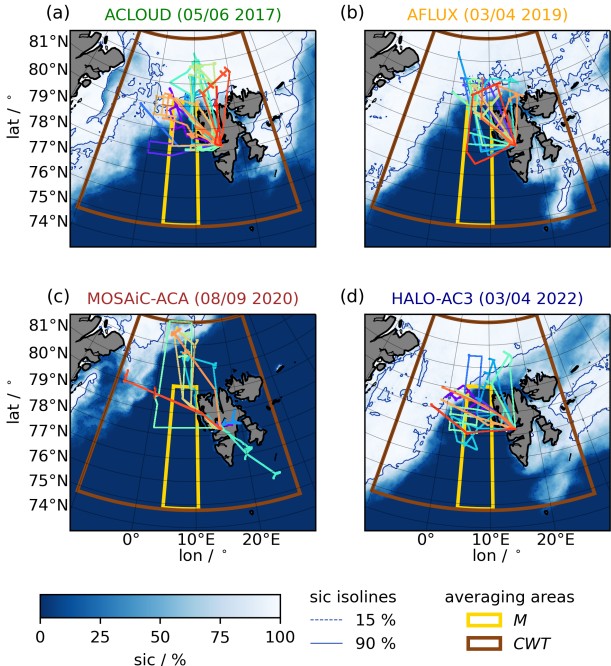

**Figure 1.** Flight tracks and sea ice concentration (sic) during the airborne campaigns ACLOUD (a), AFLUX (b), MOSAiC-ACA (c), and HALO-(AC)³ (d). The indicated areas highlight the regions over which we calculate the marine cold air outbreak index ($M$; yellow) and determine the Circulation Weather Type ($CWT$; brown).

## 2 Data and methods

All four airborne campaigns were based in Longyearbyen, Svalbard, and included various flights focusing on the Fram Strait area with varying sea ice conditions during the different campaigns (Fig. 1, Table 1). This study uses only measurements with flight altitude above 2 km and omits measurements over land due to the complex topography. We differentiate between open water (sea ice concentration (sic) $< 15\%$) and sea ice (sic $> 90\%$) using the second Advanced Microwave Scanning Radiometer (AMSR2) sea ice concentration dataset (version 5.4). In total, 82 h flight time corresponding to a distance exceeding 25,000 km are analyzed with the majority (64 %) over open ocean.

CloudSat's Cloud Profiling Radar (CPR) and MiRAC are both downlooking W-band radars operating at 94 GHz. We focus on the measurements of the equivalent radar reflectivity factor $Z$ from MiRAC ($Z_M$) and CPR ($Z_C$). The fraction of $Z$ signals in the measurement period with height is also called hydrometeor fraction and hereafter referred to as cloud fraction $CF$.

Note, that attenuation by supercooled liquid layers and precipitation affects the downward looking observations of both instruments the same way. While dry air negligibly attenuates $Z$ at 94 GHz, atmospheric water vapor and hydrometeors can significantly attenuate $Z$. In the dry Arctic, nonetheless, this attenuation is assumed small. With a total column water vapor amount of 15 kg m$^{-2}$, which is relatively high for the Arctic, a two way attenuation below 1 dBZ would occur (Kneifel et al.,





**Table 1.** Flight hours over several surfaces and covered distances during the analyzed flights of the four Polar 5 campaigns. Sea ice concentrations below 15 % and above 90 % represent open water and sea ice.

| campaign | start | end | year | all / h | sea ice / h | open water / h | distance / km |
|---|---|---|---|---|---|---|---|
| ACLOUD | 22 May | 28 June | 2017 | 22 | 8 | 10 | 7,016 |
| AFLUX | 20 March | 15 April | 2019 | 13 | 2 | 9 | 4,134 |
| MOSAiC-ACA | 27 August | 17 September | 2020 | 15 | 1 | 13 | 4,761 |
| HALO-(AC)[3] | 05 March | 15 April | 2022 | 31 | 6 | 22 | 9,803 |
| sum | | | | 82 | 17 | 53 | 25,714 |

2015). A 500 m thick cloud with a liquid water path of $100\,\mathrm{g\,m^{-2}}$ would weaken $Z$ by less than 0.6 dBZ (Stephens et al., 2002). Note that unlike the CPR, MiRAC does not suffer from atmospheric attenuation by hydrometeors above Polar 5 flight altitude, which is mostly around 3 km.

## 2.1 Spaceborne CloudSat Cloud Profiling Radar (CPR)

The CloudSat satellite orbit reaches up to 82.5° latitude and provides the only domain-wide vertically resolved satellite observations sensitive to clouds, light precipitation, and snow in the Arctic region (Liu, 2008; Kulie and Bennartz, 2009; Palerme et al., 2014). The CPR (Table 2 for a list of specifications) is a pulsed radar and the pulse width results in a range resolution of 480 m (Stephens et al., 2002; Tanelli et al., 2008). The antenna half power beam width and flight altitude cause a latitude-dependent across-track resolution of 1,320 to 1,380 m, which is 1,375 m particularly around Svalbard (Fig. 2; Table 2). Due to the integration time, the distance between two adjacent measurement center points $d$ is 1,090±10 m, which again depends slightly on latitude (Tanelli et al., 2008). As a result of the instantaneous footprint and the integration time, the effective along-track CloudSat resolution $res$ during the campaigns is close to 1,780 m. In 2006, the CPR sensitivity was close to -30 dBZ and was supposed to stay at least close to -26 dBZ (Stephens et al., 2002; Tanelli et al., 2008; Stephens et al., 2008). Due to ground clutter, CloudSat likely overestimates low-level cloud occurrences below 0.5 and 1 km height over ocean and land/sea ice, respectively (Marchand, 2018; Maahn et al., 2014; Mioche et al., 2015; Lamer et al., 2020).

We analyze CPR data from the '2B-Geoprof' product version 5 (Marchand, 2018) over four underflights of Polar 5 below CloudSat (Table 3) following Blanchard et al. (2014) and Lamer et al. (2020). This product contains $Z$ and a CPR cloud mask, which assigns a value for the cloud detection probability, every 240 m in height and 1 km along track. The '2B-Geoprof' product hereby oversamples the return power by a factor of 2. The altitudes of the CloudSat range bins are slightly variable over time. For the analysis, the data are mapped to a constant grid with a grid size of 240 m by selecting the nearest neighbor. For the cloud mask, we settle for a given confidence value of 20 or higher following Lamer et al. (2020). This means that all range gates with lower values are considered as cloud free filtering ground clutter and very weak signals (Marchand, 2018). Furthermore, only $Z_\mathrm{C}$ larger than -27 dBZ are considered as cloud signals. This threshold is in accordance with the one applied by the CPR cloud mask above the blind zone (Marchand, 2018).



**Table 2.** Specifications of the Cloud Profiling Radar (CPR) on CloudSat and the airborne radar MiRAC, which are illustrated in Fig. 2.

| parameter | CPR | MiRAC |
|---|---|---|
| flight altitude | 730 km | 3.09 km |
| flight speed | 7,000 ms$^{-1}$ | 87 ms$^{-1}$ |
| frequency | 94 GHz | 94 GHz |
| integration time | 0.16 s | 1 s |
| pulse width | $3.3 \cdot 10^{-6}$ s | - |
| range resolution | 480 m | 4.5–27 m |
| across-track resolution | 1,320–1,380 m | 460 m |
| half power beam width | <0.12° | 0.85° |
| footprint radius $r$ | 688 m | 23 m |
| distance between two measurement center points $d$ | 1,093 m | 87 m |
| effective along-track resolution $res$ | 1,780 m | 110 m |

**Table 3.** Specifications of the four underflights of Polar 5 below CloudSat.

| case | flight segment | date | start time / UTC | end time / UTC | space between platforms at crossing / m | time difference between platforms / min | |
|---|---|---|---|---|---|---|---|
| | | | | | | start | end |
| 1 | ACLOUD: RF06_hl03 | 27 May 2017 | 09:48:36 | 10:44:27 | 1,208 | 38.7 | 17.8 |
| 2 | ACLOUD: RF11_hl02 | 02 June 2017 | 09:40:39 | 09:56:59 | 1,047 | 7.0 | 8.4 |
| 3 | AFLUX: RF09_hl03 | 01 April 2019 | 09:35:50 | 10:11:59 | 1,031 | 16.6 | 20.1 |
| 4 | AFLUX: RF13_hl03 | 07 April 2019 | 08:23:35 | 08:49:59 | 802 | 5.0 | 21.1 |

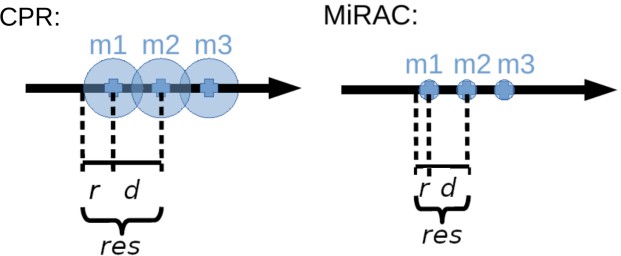

**Figure 2.** Sketch of the horizontal resolution of the radar on CloudSat (CPR; left) and airborne radar MiRAC (right). The individual measurement center positions (m1, m2, and m3) are indicated by blue crosses and each footprint by blue circles. $r$ is the radius of the footprint, $d$ the distance between two measurements, and $res$ the effective along-track resolution. For better illustration, MiRAC is scaled up by a factor of 10.





## 2.2 Airborne

MiRAC is a frequency-modulated continuous wave (FMCW) radar and operates at the same frequency (94 GHz) as CloudSat (Table 2 for list of specifications). Its sensitivity and vertical resolution depends on the chirp settings. During the campaigns the settings were such that the detection limit mostly reached below -40 dBZ (Mech et al., 2019). The vertical resolution is
4.5 m close to the aircraft and at most 27 m (Mech et al., 2019). During the processing, the vertical resolution of all flights is interpolated to 5 m. Considering the beam width, the radius of the beam's footprint at the surface is 23 m for the average flight altitude of about 3 km (Fig. 2). Due to the aircraft speed and temporal resolution of roughly 1 s, each measurement covers about 110 m. Hence, $res$ of MiRAC is roughly 16 times higher than the one of CPR. $Z_{\mathrm{M}}$ is not investigated inside the lowest 150 m of the atmosphere due to surface type-dependent ground clutter (Mech et al., 2019) and is linearly interpolated to a temporal
resolution of 1 s. This study only accounts for measurements along straight flight segments over ocean that exceed a flight altitude of 2 km (Risse et al., 2022).

The Airborne Mobile Aerosol lidar (AMALi; Stachlewska et al., 2010) also operated on Polar 5 is used to assess the cloud situation during the four underflights. It measures profiles of backscattered intensities at 532 (parallel and perpendicular polarized) and 355 nm (not polarized). After averaging these profiles over 5 s and correcting them for the background signal and
a drift, the attenuated backscatter coefficient is calculated (Ehrlich et al., 2019). By determining the highest altitude of consecutive heights that exceed the backscatter coefficient of a cloud-free section, the cloud top height is obtained with a vertical resolution of 7.5 m and a horizontal resolution of 375 m (Kulla et al., 2021a; Kulla et al., 2021b). For this study, we accessed all airborne data via the ac3airborne module and he therein included intake catalogs (Mech et al., 2022).

## 2.3 Forward-simulation methodology

This section summarizes the steps applied to convert the finer resolved and more sensitive MiRAC to CloudSat radar reflectivities.

I. Along-track convolution: We calculate a moving time average over thirteen profiles, which represent the number of MiRAC along-track bins ($res$ of 110 m) within the CloudSat footprint (1,375 m), and consider an along-track weighting function that imitates the antenna pattern by a symmetrical Gaussian distribution covering the CloudSat footprint (Lamer
et al., 2020).

II. Along-track integration: Here, the integration distance of CloudSat (1,093 m) is considered by calculating an arithmetic mean over all convoluted profiles within the integration distance. For the underflights (Sect. 2.1), we assign to every CloudSat observation the averaged profile that resembles the distance between CloudSat and the location where Polar 5 and CloudSat are closest (crossing location) best. For the statistical assessment over all campaigns, a profile is selected
every 1,093 m.

III. Along-range convolution: The range resolution of the $Z_{\mathrm{C}}$ and $Z_{\mathrm{M}}$ product is 240 m (Sect. 2.1) and 5 m (Sect. 2.2), respectively. To account for the pulse-limited range resolution of CloudSat, we average the convoluted observations



from the previous step by applying a running mean with a symmetrical, 960 m long range-weighting function following
Lamer et al. (2020). The range-weighting function is modeled with the help of a Gaussian distribution that produces
a surface clutter echo profile similar to that observed by the CloudSat CPR postlaunch. The distribution spans twice
CloudSat's range resolution, i.e., 960 m, and thereby simulates ground clutter even more realistically, since the weight
of signals far away from the center is tiny. Afterwards, we select $Z$ values for every 240 m to mimic the digitization of
CloudSat.

IV. Sensitivity threshold: To obtain the fully forward-simulated equivalent radar reflectivities $Z_{\mathrm{sim}}$, we apply a sensitivity
threshold of -27 dBZ to eliminate signals that fall below the CloudSat sensitivity due to averaging over cloudy and cloud
free bins (i.e., partial beam filling).

## 3 Meteorological conditions during airborne campaigns

The flights during the four campaigns (Fig. 1) span a range of meteorological conditions. We characterize and relate cloud
occurrence to these conditions by determining the daily marine cold air outbreak index ($M$; Papritz et al., 2015) and the
Circulation Weather Type ($CWT$; Akkermans et al., 2012) from ERA5 reanalysis data provided on pressure levels (Hersbach
et al., 2020).

### 3.1 Marine cold air outbreak index ($M$)

Following Papritz et al. (2015) and Kolstad (2017), $M$ is defined as the difference between potential temperatures $\theta$ at the
surface and 850 hPa altitude for each grid point over water:

$$M = \theta_{surf} - \theta_{850\,hPa}. \tag{1}$$

For a more robust estimate, daily $M$ values are averaged for the Fram Straight area (Fig. 1, yellow). A $M$ below -8 K classifies
a warm period, whereas a $M$ above 0 K identifies Cold Air Outbreaks (CAOs) following Knudsen et al. (2018). CAOs typically
occur when cold air masses form over the central Arctic ice and move southward over the warm open ocean where they quickly
saturate. Over the open water, cloud streets evolve, which grow in the vertical and horizontal directions with distance to the ice
edge until they form convective cells. The heat release from the ocean enhances turbulence that deepens the cloud layer with
time (Etling and Brown, 1993; Atkinson and Wu Zhang, 1996; Brümmer, 1999). Air-mass transformation during CAO still
poses many questions requiring detailed measurements for testing high-resolution modeling.

In total, 63 % of the analyzed measurements were taken during CAOs, 32 % in neutral, and 5 % in warm conditions (Fig. 3).
Note that the sampling is effected by weather conditions suitable for flying. During warm conditions, a thick, continuous, low
cloud layer often hinders Polar 5's take off and landing at Longyearbyen airport (Svalbard). Therefore, warm periods do not
appear representative.

ACLOUD (early summer) includes frequent CAO events and less warm periods (Fig. 3). During AFLUX and HALO-
(AC)³, both taking place in early spring, CAO occurrences clearly dominate the analyzed flights. Conversely, neutral conditions



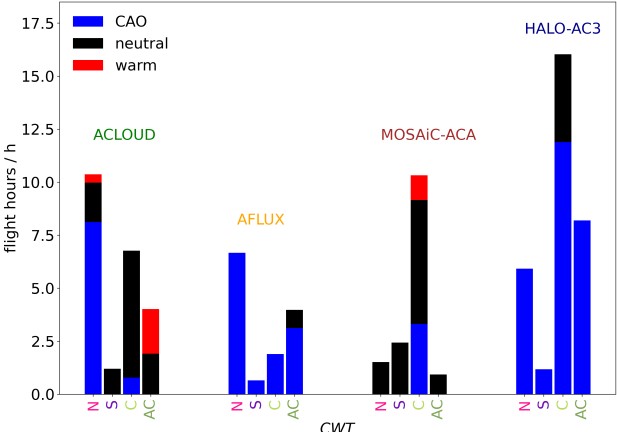

**Figure 3.** Analyzed flight hours during the different Circulation Weather Types ($CWT$s) for each campaign. N, S, C, and AC stand for northerly, southerly, cyclonic and anticyclonic flow, respectively. Each $CWT$ class is divided into the occurrence of the marine cold air outbreak index ($M$): warm periods (red), neutral periods (black) and cold air outbreaks (CAOs; blue).

($-8\,\mathrm{K} < M < 0\,\mathrm{K}$) dominate MOSAiC-ACA, which was conducted in autumn. Only twice as much flight time was conducted during CAOs than during warm periods in this autumn campaign.

## 3.2 Circulation Weather Type ($CWT$)

Several approaches to classify synoptic situations by their large-scale atmospheric circulation into $CWT$ exist, e.g., by analyzing the areal average of the vorticity, strength, and direction of the geostrophic flow. We follow Akkermans et al. (2012) and use the Jenkinson-Collison classification, which comprises eight directional classes (N, NE, E, SE, S, SW, W, and NW) and two vorticity regimes (cyclonic (C) and anticyclonic (AC); Philipp et al., 2016). For a representative assessment, the $CWT$ is calculated from the geopotential height at 850 hPa over a larger area (Fig. 1, brown).

In general, the flow is directed in meridional direction and flow directions W and E do not occur during the analyzed flights. Thus, the main flow directions are S, N, C, and AC and classes in between are assigned to the neighboring main direction following von Lerber et al. (2022). In total, 0.6 h were flown during NE and NW flows and 1.8 h during SE and SW conditions, thus, they contribute by less than one percent.

During all campaigns, northerly (30 %) and cyclonic (43 %) flows dominate whereas southerly winds appear rarely with 7 % of the analyzed flight time (Fig. 3). Least northerly winds occurred during the MOSAiC-ACA campaign (autumn). The amount of cyclonal flow is with 1.9 h lowest during AFLUX. For AFLUX and HALO-(AC)[3] (both early spring), the primary difference in the synoptic situation is the main flow type, being northern and cyclonic during AFLUX and HALO-(AC)[3], respectively. Northerly winds generally implicate CAOs except for MOSAiC-ACA, during which the number of CAOs is in general low.





Cyclonic conditions frequently include CAOs during the spring campaigns AFLUX and HALO-(AC)³, while they are less frequent (< 30 %) during MOSAiC-ACA and ACLOUD (early summer).

## 4   CloudSat underflights

Four CloudSat underflights (Table 3) were performed in the vicinity of Svalbard (Fig. A1) during ACLOUD and AFLUX
lasting about 35 min each. $Z_\mathrm{M}$ time series resolve the fine structures inside the clouds (Fig. 4a) and demonstrate that the cloud conditions during the underflights differ significantly. The clouds during case 1 and 3 reach altitudes up to more than 2 km and show light precipitation as evidenced by reflectivities in the lowest range gate. During case 2, a thin cloud layer with virga and $Z_\mathrm{M}$ below -20 dBZ appears below 1 km. Case 4 is mostly cloud free and exhibits only one small non-precipitating cloud below 2 km. During all cases, CloudSat observes no additional clouds at higher levels (not shown). Hence, no attenuation occurs
through high clouds. The cloud top heights obtained from the AMALi lidar and MiRAC measurements generally agree well. Exceptions occur at very low levels when the lidar likely detects a thin supercooled layer which is even beyond the sensitivity limit of MiRAC.

The horizontal cloud cover from the MiRAC observations during all underflights is 74 % and 45 % of the cloud tops fall within the lowest kilometer. $Z_\mathrm{M}$ ranges from -31 to 8 dBZ (Fig. 5a). Precipitation, which is hereafter defined as $Z$ larger -
5 dBZ (Maahn et al., 2014), is rare. The vertically resolved $Z_\mathrm{M}$ distribution (Fig. 5g) reveals that precipitation is confined to below 750 m height. $Z_\mathrm{M}$ is most frequent at -15 dBZ due to signals between 0.15 and 1.2 km height that are mainly observed during case 1.

The mean $CF$ profile of MiRAC ($CF_\mathrm{M}$; Fig. 6a, MiRAC) over all underflights shows almost no clouds above 2 km, on average 15 % clouds between 1.5 and 2 km height, and an increase up to 40 % between 1.5 and 1 km height. At around 750 m,
$CF_\mathrm{M}$ maximizes with 53 % over less than 500 m mainly due to the cloud layer captured during case 2.

### 4.1   Effect of the forward simulation

At first, we illustrate how the different processing steps (Sect. 2.3) change the radar reflectivities when converting the MiRAC measurements ($Z_\mathrm{M}$) to those that would be observed by CloudSat ($Z_\mathrm{sim}$):

    I. The along-track convolution (Fig. 4b) is independent of height and smooths hydrometeor related signals in the horizontal.
230        Therefore, especially broken cloud fields with small gaps are combined into clouds with larger horizontal extent. Also isolated reflectivities, such as during case 4 at altitudes below 200 m, become visible by smearing over a larger distance. The occurrence of very low-level clouds is confirmed by the lidar, which, however, mostly fall even below the sensitivity limit of MiRAC. Note, that at cloud boundaries, $Z$ often declines below the sensitivity threshold of -27 dBZ (Fig. 5b). Compared to the original $Z_\mathrm{M}$ distribution, which has its maximum at -15 dBZ (Fig. 5a), the distribution becomes bimodal
235        (Fig. 5b).



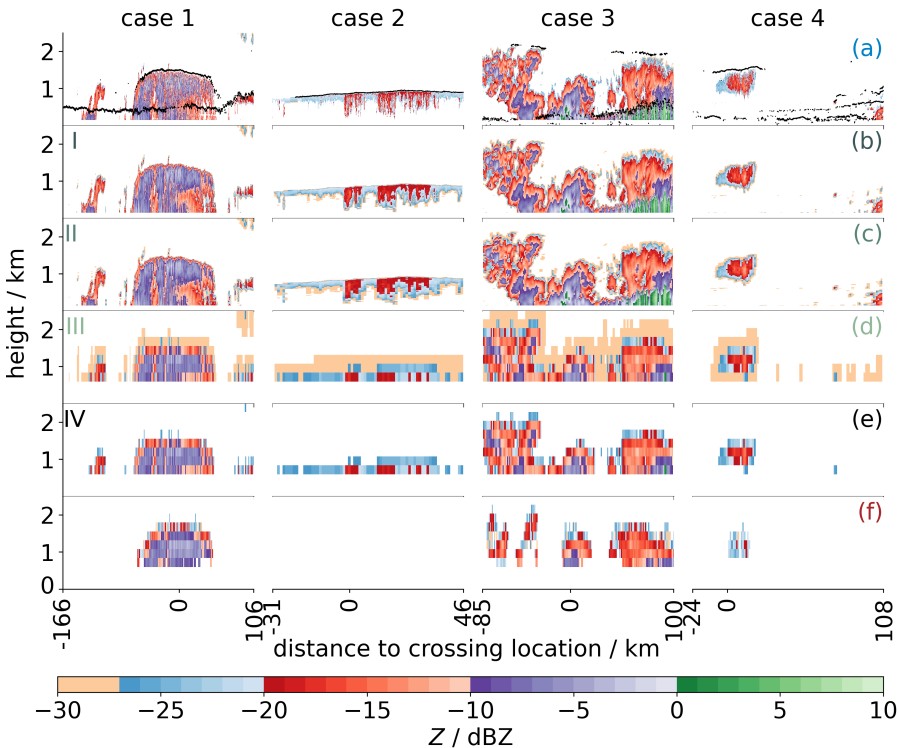

**Figure 4.** Profiles of the equivalent radar reflectivity $Z$ during the four underflights of Polar 5 below CloudSat (columns) as obtained from the airborne radar MiRAC ($Z_{\mathrm{M}}$; a), after along-track convolution (I; b), additional along-track integration (II; c), further along-track convolution (III; d) and after applying a sensitivity threshold of -27 dBZ ($Z_{\mathrm{sim}}$; IV; e). The CloudSat observations $Z_{\mathrm{C}}$ (f) are filtered by the CPR cloud mask. In addition, the cloud top height derived by the airborne lidar AMALi (a; black dots) are shown.

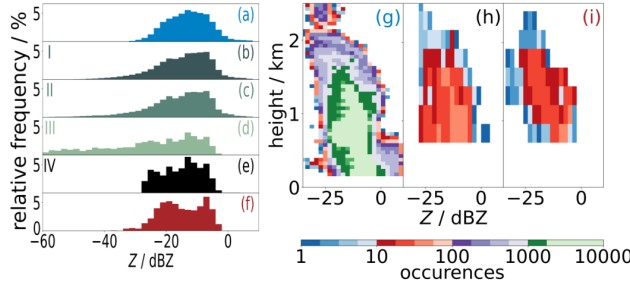

**Figure 5.** Histogram (a–f) and contoured frequency by altitude diagram (CFAD; g–i) of the equivalent radar reflectivity $Z$ over four underflights of Polar 5 below CloudSat. Histograms display the original MiRAC ($Z_{\mathrm{M}}$; a) and CloudSat data ($Z_{\mathrm{C}}$; f), and forward-simulated data obtained after each processing step (I – IV; Sect. 2.3). CFADs show $Z_{\mathrm{M}}$ (g), the completely forward-simulated data ($Z_{\mathrm{sim}}$; h), and $Z_{\mathrm{C}}$ (i). The color coding of the labels is equivalent to Fig. 4. The size of the bins equals 2 dBZ.





II. The along-track integration (Fig. 4c) broadens and smears cloud structures in the horizontal, e.g., cloud gaps clearly shrink during case 1. The bimodality of Fig. 4b strengthens and the distribution now has a global maximum at -8 dBZ and a local maximum at -20 dBZ (Fig. 5c).

III. After the along-range convolution (Fig. 4d), the coarser vertical resolution displays less fine cloud structures, stretches clouds in the vertical, and hence increases cloud top heights. To illustrate this in detail: the range-weighting function averages $Z$ at cloud top over a range of $\pm 480$ m. Thus, cloud free conditions above cloud top now show a non-negligible radar reflectivity moving the cloud top upwards. Similar to the along-track averaging, $Z$ decreases drastically even below -60 dBZ (Fig. 5d).

IV. The sensitivity threshold (Fig. 4e compared to c) reduces the number of signals along cloud boundaries and of whole clusters during case 4 and enlarges cloud gaps.

The lower resolution and sensitivity of $Z_{\mathrm{sim}}$ do not change the contoured frequency by altitude diagram (CFAD) compared to $Z_{\mathrm{M}}$ at the most frequented regions between -15 dBZ and -8 dBZ (Fig. 5g, h). However, the forward simulations decrease the number of $Z_{\mathrm{sim}}$ above 2.25 km and increase the number of $Z_{\mathrm{sim}}$ smaller 22 dBZ below 960 m height. $Z_{\mathrm{sim}}$ do not resolve the lowest 720 m and hence almost no precipitation.

In a second step, we analyze how the mean $CF$ profile averaged over all underflights changes by each processing step:

I. The along-track convolution (Fig. 6b, MiRAC compared to I) has no effect on $CF$ above 1.5 km, increases $CF$ by roughly 3 pp between 1 and 1.5 km and by 10 pp below. Near the surface, the overestimation of $CF$ increases from 7 to 25 % of $CF_{\mathrm{M}}$. The change in $CF$ due to along-track averaging depends on how many individual clouds are encountered.

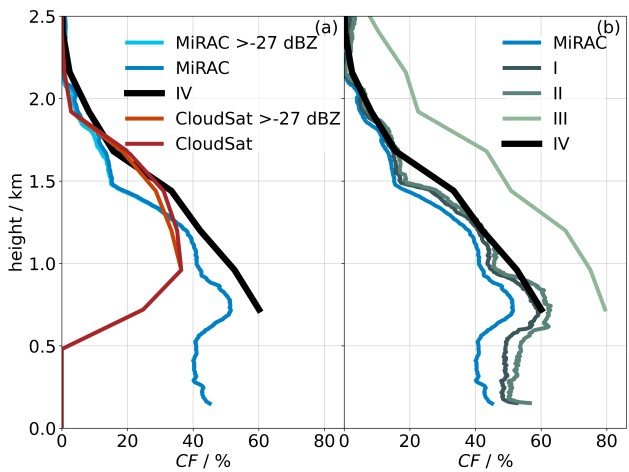

**Figure 6.** Cloud fraction $CF$ profiles over four underflights of Polar 5 below CloudSat. Original and completely forward-simulated (IV) profiles are displayed in (a). The effect of each processing step (I – IV; Sect. 2.3) on $CF$ is illustrated in (b). The color coding is equivalent to Fig. 4.





A larger number of clouds with short gaps in between, e.g., across cloud streets, favors more horizontal cloud stretching and thus increases $CF$ more. The clouds in case 3 enhance $CF$ over all heights, whereas the precipitation and virga in case 1 and 2 intensify the low-layer $CF$ increase.

II. The along-track integration (Fig. 6b, I compared to II) acts in the same way as the along-track convolution but with a smaller effect. An additional increase of $CF$ occurs, which is strongest below 1 km but less than 3 pp.

III. The range convolution (Fig. 6b, II compared to III) shifts the $CF$ profile up by about 480 m due to cloud top stretching as $Z$ are averaged over hydrometeor free areas. Below 1 km, $CF$ increases additionally, thus by around 30 pp in total. Here, the effect of the range-weighting function (Sect. 2.3) spanning $\pm$ 480 m is evident. At 720 m, $Z$ between 240 m and 1.2 km affect $CF$ after the range convolution. Hence, the range-weighted signals from non-precipitating low-level clouds might reach the lowest level. As we do not explicitly model a surface reflection signal, this weighting is also the reason why $CF_{\mathrm{sim}}$ can only be calculated down to 720 m.

IV. $CF$ after applying the sensitivity threshold ($CF_{\mathrm{sim}}$; Fig. 6b, III compared to IV) reduces by 25 pp particularly just above 1.5 km. Most cloud tops are directly below 1.5 km, where the gradient of the $CF_{\mathrm{M}}$ profile is strongest. After cloud stretching, $Z$ at the cloud tops are very small and often fall below the threshold. Thus, the effect of the threshold is predominant at 1.9 km, i.e., 480 m above the layer with most cloud tops. The sensitivity threshold reduces the cloud top height overestimation and leads to a net overshooting of about 240 m compared to $CF_{\mathrm{M}}$. Note that close to 1.9 km, some $Z_{\mathrm{M}}$ already fall below the threshold (Fig. 6a, MiRAC $> -27$ dBZ compared to MiRAC).

In summary, changes in sensitivity and resolution (Fig. 6a, MiRAC compared to IV) enhance $CF_{\mathrm{sim}}$ compared to $CF_{\mathrm{M}}$ strongest below 1.5 km, i.e., 11 pp at 720 m that is 25 % of $CF_{\mathrm{M}}$.

## 4.2 Evaluation of the forward simulation

Can we use $Z_{\mathrm{sim}}$ as a proxy for $Z_{\mathrm{C}}$ and thereby expand the analysis period over all campaigns? A comparison of $Z_{\mathrm{sim}}$, $Z_{\mathrm{C}}$ and the corresponding $CF$ profiles shall answer this question and detect measurement biases between MiRAC and CPR. Note that differences in the observed cloud fields can arise due to the time and location shifts between the two radars. The highly spatially and temporally variable clouds (Fig. 4) do not allow to project clouds in space and time as done by Gayet et al. (2009) for extended ice clouds.

$Z_{\mathrm{sim}}$ and $Z_{\mathrm{C}}$ agree well (Fig. 4e, f) though a lower number of signals is measured by CloudSat. This is most striking in case 2, when CloudSat detects no clouds. Note that in the raw CloudSat data, i.e., without applying the cloud mask, weak $Z_{\mathrm{C}}$ appear at 1 km height during case 2 (not shown). However, the mask attributes these signals to ground clutter and generally filters all signals below 720 m. The fact that MiRAC measurements for case 1 and 3 evidence significant hydrometeor occurrence below 1 km demonstrates that the cloud mask is too strict as pointed out in Lamer et al. (2020) (see their Fig. 1). Compared to the horizontal cloud cover from MiRAC, the one observed by CloudSat thus reduces by a factor of 2. Ground clutter could cause artificial echos in the lower layers, if the CPR mask is too gentle. In this case, ground clutter would enhance $Z_{\mathrm{C}}$ but not affect



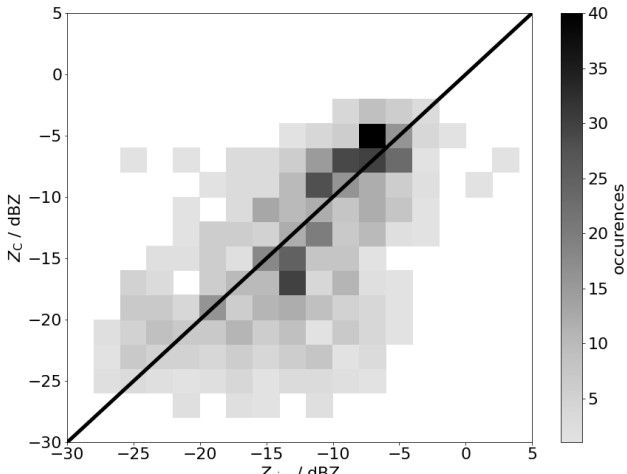

**Figure 7.** Comparison of the equivalent radar reflectivity obtained from forward simulations $Z_{\mathrm{sim}}$ and CloudSat $Z_{\mathrm{C}}$ over four underflights of Polar 5 below CloudSat. The bin size equals 2 dBZ.

$Z_{\mathrm{sim}}$, which is not dominant during the underflights (Fig. 7). Furthermore, CloudSat does not detect low $Z_{\mathrm{M}}$ and thus shows separate clouds instead of a continuous cloud layer during case 3.

We investigate the realism of the forward simulation by directly comparing $Z_{\mathrm{sim}}$ and $Z_{\mathrm{C}}$ for each pixel (Fig. 7). $Z_{\mathrm{C}}$ are on average 2 dB lower than $Z_{\mathrm{sim}}$ for times when both instruments measured a signal. This bias is in the same range (1–2 dB) as

found by Protat et al. (2009). They processed the airborne $Z$ the same way we do, but used a threshold of -29 dBZ. Note that they achieve a better matching of air- and spaceborne $Z$, because they only analyze extended non-precipitating ice clouds and minimize the time and spatial lag between the measurements to below 10 min and a few hundred meters. Thus, the $RMSE$ (5.5 dB) and standard deviation (5.17 dB) of our data, which is twice the value claimed by Protat et al. (2009) (2-3 dB), are larger. However, the highly variable low clouds that are observed by each instrument differ due to the time shift and location

mismatch of the platforms, thus $Z_{\mathrm{C}} - Z_{\mathrm{sim}}$ (Fig. A2) is dependent on the distance to the underflight (time shift) and on the distance between both platforms (location shift). For measurements that are obtained within 30 km around the crossing location and when Polar 5 and CloudSat are closer than 35 km, the bias and standard deviation decrease to -0.37 and 3.18 dB, respectively.

Having shown the good agreement between forward-simulated and measured reflectivities, we now focus on the vertical

cloud fraction profile. Above 1.5 km, the profiles agree very well, $CF_{\mathrm{C}}$ deviates from the $CF_{\mathrm{sim}}$ profile by less than 5 pp (Fig. 6a, IV compared to CloudSat). This agreement worsens for lower altitudes, $CF_{\mathrm{C}}$ is lower by 36 (16 pp) at 720 (960 m) height, which is 60 (31 %) of $CF_{\mathrm{sim}}$. This is consistent with the omission of signals by CloudSat due to an too aggressive cloud mask as discussed above. In summary, the comparison demonstrates that $Z_{\mathrm{sim}}$ can be used as a good proxy for $Z_{\mathrm{C}}$ above 1.5 km but that care has to be taken below especially in the blind zone. This holds particularly for the maximum in $CF_{\mathrm{M}}$ of 50 % measured

at 720 m that $CF_{\mathrm{sim}}$ overestimates but CloudSat observations strongly underestimate.





## 5 Evaluation of CloudSat limitations during campaigns

Synthetic CloudSat reflectivity profiles $Z_{\text{sim}}$ are generated from the MiRAC observations carried out over the four campaigns (Tab. 1) and serve as a base for assessing CloudSat's limitations. We first investigate the effect of these limitations on cloud fraction profiles (Sect. 5.1) derived from $Z_{\text{sim}}$ and specify the drivers for differences between forward simulations and "truth".

Furthermore, we analyze how much multilayer clouds (Sect. 5.2) and precipitation (Sect. 5.3) are affected. Note, that these campaign measurements can not be considered as a climatology, however, they provide unique data and insights into Arctic low-level clouds.

### 5.1 Cloud fraction profiles

Averaged over the four campaigns, the observed vertical cloud fraction profile $CF_{\text{M}}$ is 12 % for altitudes above 1.5 km and in-
creases to 40 % towards the surface (Fig. 8a). The increase is strongest between 1.5 and 0.6 km and the high values at MiRAC's lowest height of 150 m indicate frequent precipitation and probably very low clouds (Griesche et al., 2021). Excluding the blind zone, we average cloud fraction between 1 and 2.5 km and assess the impact of the different forward simulation steps (Sect. 2.3). $CF$ increases by 6 pp due to CloudSat's along track convolution and integration (Fig. 8a, MiRAC compared to II), i.e., horizontal resolution, increases by 12 pp due to its range resolution (II compared to III) and decreases by 10 pp due
to its sensitivity (III compared to IV). Vertically resolved, maximum effects of +25 pp (horizontal resolution), +20 pp (range resolution) and -30 pp (sensitivity) occur. The horizontal cloud cover between 1 and 2.5 km reduces only by 5 pp to 34 % during the forward simulation.

Mean $CF_{\text{M}}$ over the lowest 2.5 km varies between 17 % during MOSAiC-ACA and 25 % during AFLUX (Fig. 8b). This is even more pronounced below 1.25 km, when $CF_{\text{M}}$ differs between 20 and 60 % and might reflect a difference between autumn
(MOSAiC-ACA) and spring (AFLUX). The profiles obtained during ACLOUD and HALO-(AC)[3] resemble the mean profile over all campaigns. The shape of the $CF_{\text{M}}$ profile varies between the campaigns, again with the largest differences between AFLUX and MOSAiC-ACA. While MOSAiC-ACA features a roughly constant vertical $CF_{\text{M}}$ of around 20 % in the lower troposphere, AFLUX has the lowest $CF_{\text{M}}$ of about 10 % at higher altitudes that strongly increases to 65 % towards the surface.

The average vertical profiles of the forward-simulated $CF_{\text{sim}}$ and measured $CF_{\text{M}}$ profiles show a similar shape (Fig. 8a
MiRAC and IV). However, the absolute difference between $CF_{\text{sim}}$ and $CF_{\text{M}}$ (Fig. 8d) reveals that $CF_{\text{sim}}$ is larger and the difference increases towards the surface. At the lowest forward-simulated height (0.72 km), $CF_{\text{sim}}$ overestimates $CF_{\text{M}}$ by 11 pp, i.e., CloudSat would overestimate cloud fraction by one third. The increasing overestimation towards the surface is evident for all campaigns (Fig. 8e) though differences of about 5 pp are evident. In particular, a peak in the overestimation at 1.5 km height occurs during ACLOUD (Fig. 8e) that might depend on differences in the cloud situation.

As already illustrated for the underflights (Sect. 4.1), CloudSat's lower vertical resolution shifts $CF_{\text{sim}}$ vertically up by roughly 240 m and stretches the clouds. In conclusion, low-level clouds are overestimated above the blind zone but the dominant hydrometeor layer below 750 m is completely missed by the blind zone. CloudSat's performance, i.e., $CF_{\text{sim}} - CF_{\text{M}}$, does not show clear differences between the campaigns performed in different seasons. This might depend on the probed cloud types,





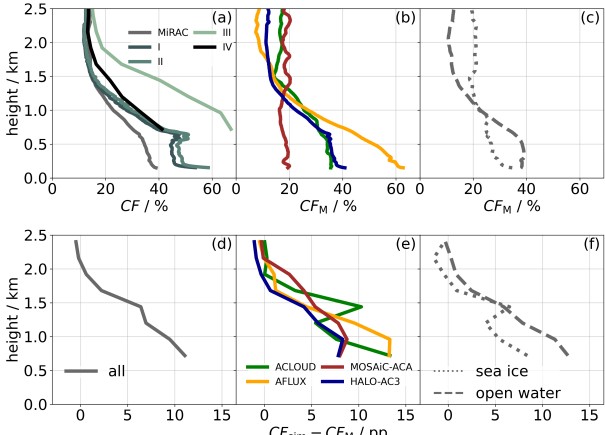

**Figure 8.** Cloud fraction profiles from the airborne radar MiRAC $CF_M$ over four campaigns (first row) and the difference compared to the forward-simulated profiles $CF_{sim}$ (second row). Profiles are averaged over all data (a, d), each campaign (b, e) and different surface covers (c,f). Sea ice concentrations below 15 % and above 90 % represent open water and sea ice. Moreover, the profiles after each processing step (I – IV; Sect. 2.3) are displayed (a).

their connection to different synoptic situations and the way they are probed. In the following, we assess the dependence of

CloudSat's performance on various parameters.

Surface cover: We analyze dissimilarities between $CF_M$ and $CF_{sim}$ over open water and sea ice (Fig. 8c, f). In general, cloud fraction profiles appear different over sea ice, where they are relatively constant with height, and over open water, where higher levels have less and the lowest kilometer has more clouds. Over sea ice, a slight increase in $CF_M$ close to the surface occurs that might be related to very low-level clouds found by Griesche et al. (2021). $CF_{sim}$ overestimates $CF_M$ especially

below 1.75 km getting stronger closer to the ground (Fig. 8f). This overestimation is more pronounced over open water than over sea ice. Over ice a second maximum in overestimation occurs at 1.5 km where the cloud fraction is discontinuous.

Cold air outbreak index: $CF$ is investigated for different $M$ classes (Sect. 3.1). We only focus on CAOs and neutral conditions as too few cases for warm conditions exist, which would not allow to draw valid conclusions. During CAOs, $CF_M$ is close to 10 % above 1.5 km height (Fig. 9a), increases linearly down to 1 km, and more slowly until it reaches 50 % close

to the surface. In contrast, $CF_M$ is with about 18 % more constant over height for neutral conditions. Then, no significant differences between sea ice and open water are visible while differences up to 20 pp occur during CAOs. The latter differences vary with height similar to the overall difference between sea ice and open water (Fig. 8c). Clearly, CAOs are responsible for the highest low-level cloud fractions with significant differences between measurements over ice and open water, where air-mass transformation changes cloud characteristics along the trajectory. The atmospheric boundary layer height increases

with distance to ice edge due to strong surface fluxes. Evaporation supports the cloud development from roll cloud streets close to the ice edge to cellular convection further downstream. $CF_{sim}$ overestimates $CF_M$ by up to 16 pp mainly during CAOs (Fig. 9c), when the coarse vertical resolution deepens the low-level cloud rolls over water, and less during neutral situations, when



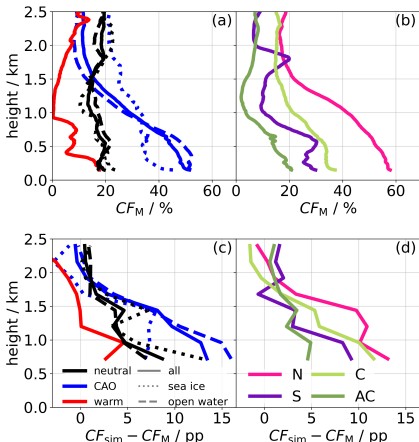

**Figure 9.** Cloud fraction profiles from the airborne radar MiRAC $CF_{\mathrm{M}}$ over four campaigns (first row) and the difference compared to the forward-simulated profiles $CF_{\mathrm{sim}}$ (second row). Profiles are averaged for different marine cold air indices ($M$; a, c; solid lines): warm period (red), neutral period (black) and cold air outbreak (CAO; blue). The data are additionally categorized into over sea ice (sea ice concentration (sic) $< 15\,\%$; dotted) and open water (sic $> 90\,\%$; dashed). Moreover, profiles are separated into Circulation Weather Types (CWT; b, d). N, S, C, and AC stand for northerly, southerly, cyclonic and anticyclonic flow, respectively.

$CF_{\mathrm{M}}$ is constant. The overestimation is strongest close to the surface, i.e., 14 and 9 pp during CAOs and neutral conditions, respectively. We speculate that the overestimation depends on cloud amount and orientation of the flight tracks in respect to

the cloud streets as this influences the number of cloud gaps over which signals are averaged.

Circulation weather type: Cyclonic flows are the most frequent $CWT$ (43 %) followed by northerly flows (Fig. 3). $CF_{\mathrm{M}}$ shows a strong dependence on $CWT$ though for all regimes the highest cloud fraction occurs in CloudSat's blind zone (Fig. 9b). Northerly flows exhibit the largest $CF_{\mathrm{M}}$. During cyclonic conditions, the shape of the profile is similar but $CF_{\mathrm{M}}$ is lower. Both flows, particularly the northerly one, favor CAOs (Fig. 3) and associated cloud rolls. During southerly winds, non-

precipitating clouds exist in altering heights. $CF_{\mathrm{M}}$ is generally lowest and often zero during anticyclonic conditions, which, however, are rare. Again $CF_{\mathrm{sim}}$ overestimates $CF_{\mathrm{M}}$ for all $CWT$ below 1.5 km. The effect is strongest during northerly conditions followed by cyclonic conditions. Although, $CF_{\mathrm{M}}$ and its overestimation by CloudSat is largest during northerly winds, both seem not directly related to $CWT$, i.e., $CF_{\mathrm{M}}$ is larger during AFLUX than HALO-(AC)[3] regardless of $CWT$ (not shown). In fact, $CF_{\mathrm{M}}$ and the difference to the synthetic profiles are sorted in the same order which implies a dependence on

the amount of cloud fraction.

In conclusion, the errors imposed by CloudSat's limitations ($CF_{\mathrm{sim}} - CF_{\mathrm{M}}$) do not show a clear dependence on the surface type, $M$ or $CWT$ but rather on cloud fraction and the shape of the profile. Significant errors only occur for clouds below 1.5 km. For the low-level cloud fraction we thus propose a simple correction in form of a linear regression: The overestimation is 5 pp at 30 % cloud fraction and increases linearly to 15 pp for a cloud fraction of 60 %. While such a correction would reduce





the overestimation of the vertically resolved cloud fraction with an residual uncertainty of about 5 pp (not shown) it has to be stressed that the blind zone neglects low-level clouds which are the most common clouds in the Arctic (Fig. 6).

## 5.2 Multilayer clouds

The radiative characteristics of multilayer and single layer cloud conditions often differ (Li et al., 2011). During ACLOUD, Mech et al. (2019) identified 38 % of the cloudy scenes to be composed of multilayer clouds that have a median thickness of 205 m. CloudSat might miss individual clouds due to its sensitivity and its coarse resolution might merge separate hydrometeor layers to a single layer (Sect. 4.1). We investigate the overall effect on the frequency of multilayer cloud occurrence by defining a profile as containing a multilayer cloud for CloudSat if a gap of at least one range gate (240 m) occurs in the $Z_{\mathrm{sim}}$ profile. For MiRAC a threshold of 90 m is used to take advantage of its finer resolution.

Averaged over all campaigns, 48 % of the cloud tops observed over all $Z_{\mathrm{M}}$ profiles belong to multilayer clouds that have a mean thickness of 347 m (single layer clouds: 762 m). During the forward simulations these multilayer clouds might merge to single layer clouds. For $Z_{\mathrm{sim}}$ only 12 % of the cloud tops belong therefore to multilayer clouds, which have a mean thickness of 527 m. The coarse resolution deepens single layer clouds by 140 m and multilayer clouds by 180 m. 43 % of the observed multilayer cloud tops are below 1 km, which is less than for single layer clouds (55 %), however, this implies that nearly every multilayer system has a layer with a cloud top below 1 km. With 48 % of all multilayer cloud tops, slightly more multilayer clouds are below 1 km for $Z_{\mathrm{sim}}$ than for $Z_{\mathrm{M}}$.

The absolute number of cloudy $Z_{\mathrm{M}}$ profiles containing medium (0.24–1.92 km) thick clouds reduces by a factor of 15 during the forward simulation (Fig. 10a, b; pink). For 240 m thick clouds, the factor is 2.3 times larger. $Z_{\mathrm{sim}}$ do not detect more than twice as many shallow than medium thick clouds (Fig. 10b; gray). Furthermore, $Z_{\mathrm{sim}}$ miss the thickest clouds. Because of the low vertical resolution, $Z_{\mathrm{sim}}$ do not resolve the lowest 720 m of the atmosphere and thus thin the thickest clouds by 240 m to 2.16 km. The ratio between the number of clouds obtained by MiRAC and the forward simulations for clouds that are thicker than 1.92 km is 70 % of the averaged ratio due to cloud stretching.

For multilayer clouds only, more 480 than 240 m thick clouds occur for $Z_{\mathrm{sim}}$ than for $Z_{\mathrm{M}}$ in relative terms (Fig. 10a, b; pink). First, shallow clouds might get stretched. Second, $Z_{\mathrm{sim}}$ might detect less thin clouds, which reduces the total number of cases. Furthermore, the absolute number of clouds over all cloud thicknesses excluding 240 m reduces by a factor of 68 during the forward simulation. The number of multilayer clouds diminishes four times as much as of all clouds due to the reduced resolution and sensitivity. $Z_{\mathrm{sim}}$ could either not detect thin, second cloud layers anymore or merge multilayer to single layer clouds. For 240 m thick clouds, the reduction factor of the number of clouds observed by MiRAC and the forward simulations is twice the average over the remaining cloud thicknesses. The detection omission is larger for shallow than deeper clouds, but decreases compared to all clouds. Thus, the detection omission of forward-simulated shallow clouds is a general shortcoming, rather than one attributed to multilayer clouds. Hence, the merging of multiple cloud layers results in the four times larger reduction factor of the number of multilayer clouds.





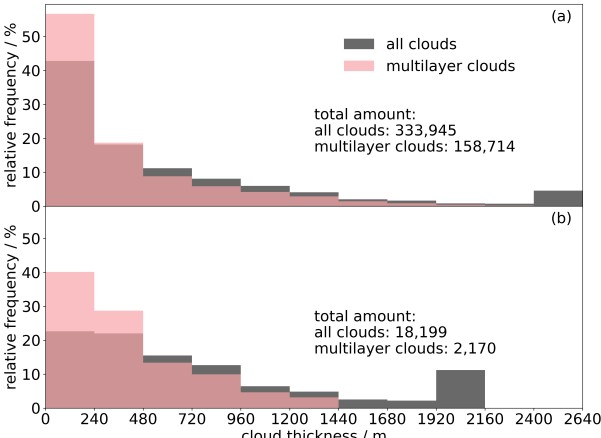

**Figure 10.** Relative frequency of occurrence for the thickness over all (gray) and multilayer clouds (pink) derived from the equivalent radar reflectivity of the airborne radar MiRAC $Z_M$ (a) and of the forward simulations $Z_{sim}$ (b) over four campaigns. Note that MiRAC resolution is much finer but binned to match the one of CloudSat. The total amount of all and multilayer clouds is displayed with each radar reflectivity profile counting as an additional cloud.

## 5.3 Precipitation

One of the most important applications of CloudSat is the derivation of snowfall in the Arctic. The Fram Strait is of particular interest as precipitation is most intense in this area (McCrystall et al., 2021) and snowfall estimates between CloudSat and regional climate models differ highly (von Lerber et al., 2022). From the '2C-Snow-Profile', Edel et al. (2020) derived a mean snowfall rate $S_C$ of 200 to $500\,mm\,yr^{-1}$ around Svalbard. However, $S_C$, which is calculated for the near-surface bin in $1.2\,km$ height that is assumed to be the lowest bin not affected by ground clutter, deviates from the surface snowfall rate $S_{surf}$. Moreover, resolution limitations (Sect. 2.1) might affect $S_C$.

We calculate snowfall rates from $Z_M$ ($S_M$) and $Z_{sim}$ ($S_{sim}$) for $Z$ larger than $-5\,dBZ$ via the $Z_e$-$S$ relation for three bullet rosettes following Maahn et al. (2014). We derive $S_M$ for all heights above $150\,m$ to avoid ground clutter contamination for MiRAC (Sect. 2.2).

First, the effect of CloudSat's resolution on the $Z_{sim}$ and $S_{sim}$ distributions is investigated at $1.2\,km$ by comparing them with the respective $Z_M$ and $S_M$ distributions. Compared to $Z_M$, the relative number of $Z_{sim}$ between -5 and 3 dBZ is larger and of stronger $Z_{sim}$ lower (Fig. 11a), i.e., CloudSat would overestimate very light snowfall and underestimate stronger snowfall. $Z$ decreases during the spatial convolution. Note that $Z_M$ might fall below the threshold for precipitation ($Z > -5\,dBZ$) during the forward simulation reducing the amount of $S_{sim}$ values. The histogram of snowfall rates shows that the number of $S_{sim}$ and $S_M$ decreases exponentially with their intensity (Fig. 11b). The relative number of $S_{sim}$ compared to $S_M$ is larger for $S_{sim}$ below $0.2\,mm\,h^{-1}$, lower for $S_{sim}$ between 0.2 and $2.0\,mm\,h^{-1}$, and comparable for $S_{sim}$ above $2\,mm\,h^{-1}$. Due to its low resolution, CloudSat would overestimate low snowfall rates by 4 pp and underestimate higher rates by 4 pp.



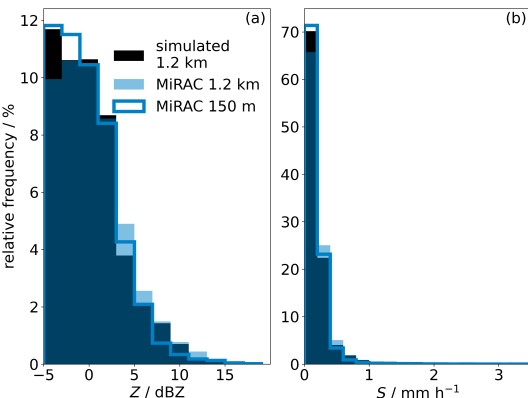

**Figure 11.** Relative frequency of occurrence of the equivalent radar reflectivity $Z$ (a) and precipitation rate $S$ (b) for $Z$ larger -5 dBZ observed by the airborne radar MiRAC in 1.2 km (blue shade) and 150 m (blue line) and by the forward simulations in 1.2 km height (black shade) over four campaigns. The size of the bins equals 2 dBZ and 0.2 mm h$^{-1}$.

We evaluate the influence of CloudSat's blind zone on its total precipitation amount $A_C$, which is the integral of the snowfall rate at a specific height over measurement time, following Maahn et al. (2014) for $Z_M$ over ocean. Over all campaigns, the total precipitation amount obtained from MiRAC ($A_M$) is 1.0 mm ($S_M$ of 111 mm yr$^{-1}$) at 1.2 km and with 2.1 mm ($S_M$ of 229 mm yr$^{-1}$) more than twice as much at 150 m. For a one year period at Ny-Ålesund, Maahn et al. (2014) found a larger $S_{surf}$ of 320 mm yr$^{-1}$ using a ground-based which might result from the choice of flight patterns that avoid storms and deep clouds. Due to its blind zone, CloudSat would underestimate $A_M$ at 150 m by 51 pp (Fig. 12) which is much stronger than for Ny-Ålesund (9 pp; Maahn et al., 2014).

To identify the $Z_M$ regime leading to the underestimation of $A_M$ caused by ClaudSat's blind zone, $A_M$ is analyzed for different reflectivity classes (Fig. 12). Closest to the ground, light precipitation ($Z_M < 10$ dBZ) is with 90 % the dominant contributor to $A_M$. These reflectivities strongly increase from 1.2 km altitude down to 500 m and less below. $Z_M$ between 10 and 20 dBZ equally contribute to $A_M$ over all heights. $Z_M$ larger 20 dBZ only occur below 400 m and contribute to $A_M$ the stronger the closer to the ground. The total precipitation amount has its maximum at 235 m height because it strongly increases below 1.2 km probably due to formation of light precipitation and slightly decreases down to 150 m due to sublimation. The increase in occurrence of higher reflectivity classes just above the maximum height is likely related to aggregation.

Light precipitation ($Z_M < 10$ dBZ) plays with 90 % a more important role than with 35 % for Ny-Ålesund (Maahn et al., 2014) while moderate and strong precipitation is much reduced. We also find a lower height of maximum precipitation (235 m vs 600 m) and less sublimation (3 pp vs 20 pp). This might be related to the generally higher latitudes and colder conditions encountered during the flights. Moreover, the recorded cloud types favor light precipitation. In particular, many CAOs occurred throughout the campaigns (Fig. 3). At least, 70 % of $A_M$ is measured during CAOs for all heights and $Z_M$ regimes. This ratio



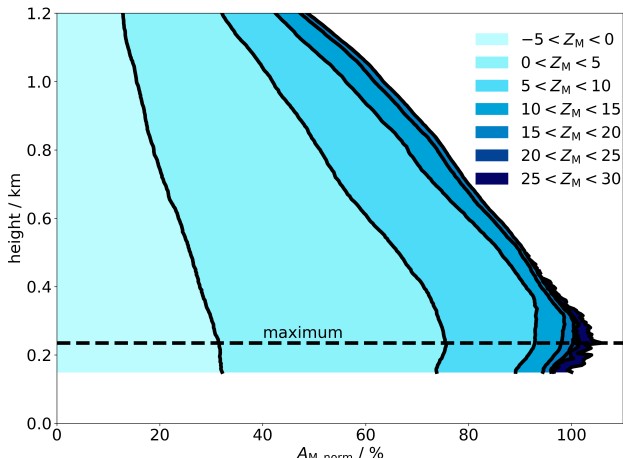

**Figure 12.** Contribution from different intervals of equivalent radar reflectivity obtained from the airborne radar MiRAC $Z_M$ to the total precipitation amount over all campaigns $A_M$ with height. $A_{M,norm}$ is the integral over the snowfall rate $S_M$ for a specific height, which is calculated for $Z_M$ larger -5 dBZ via the $Z_e$-$S$ relation for three bullet rosette following Maahn et al. (2014), normalized by $A_M$ at 150 m that is the nearest surface bin not affected by ground clutter. The dashed line at 235 m marks the height of maximal $A_{M,norm}$.

is higher in lower altitudes down to 250 m. During CAOs, the number of $Z_M$ larger 5 dBZ is that low that these $Z_M$ do not
enhance $A_M$. In summary, CAOs produce mainly light precipitation dominating $A_M$.

## 6    Conclusions and outlook

Many studies use CloudSat observations to investigate Arctic clouds (Zygmuntowska et al., 2012; Liu et al., 2012; Mioche et al., 2015) and snowfall (von Lerber et al., 2022). However, CloudSat CPR has a blind zone of about 1 km, a coarse spatial resolution, and a limited sensitivity, which impact its usefulness in the assessment of warm marine boundary layer clouds and
precipitation (Lamer et al., 2020). Our study extends this investigation for the Arctic using spatially fine resolved airborne radar reflectivity measurements by MiRAC obtained during four campaigns that took place over different seasons.

The measurements, which cover more than 25,000 km, are used to forward simulate CloudSat measurements. During four underflights, these forward-simulated and CloudSat radar reflectivities agree within 2 dB, thus the forward simulations proxy CloudSat well. The cloud fraction obtained by MiRAC over all campaigns is on average 30 % with lower values of about 15 %
at 2.5 km and a maximum of 40 % close to the ground. CloudSat's limitations increase the forward-simulated cloud fraction at 720 m by 11 pp, which is 33 % of the MiRAC cloud fraction. However, there are compensating effects at play: CloudSat's horizontal resolution increases the cloud fraction by a maximum of 25 pp, its range resolution by a maximum of 20 pp and its sensitivity decreases the cloud fraction by a maximum of 30 pp. The lower spatial resolution fills cloud gaps, stretches clouds by 240 m at cloud top/bottom, and hence increases the cloud fraction of the forward-simulated observations the stronger
the closer to the ground. Our finding that MiRAC and CloudSat radar reflectivities differ substantially below 1.5 km supports



the conclusion of Lamer et al. (2020) that the CPR cloud mask might be too restrictive such that airborne remote sensing is necessary to resolve fine cloud structures and the lowest kilometer of the atmosphere.

We aimed to identify the drivers for CloudSat's over-/underestimations: Less discrepancies between the forward-simulated and MiRAC cloud fraction occurred over sea ice than over open water. The forward simulations overestimate the MiRAC cloud

fraction with 16 pp strongest over water during cold air outbreaks mostly due to cloud top stretching. Northerly flows, mainly connected with CAOs, show the highest low-level cloud fraction and overestimation by CloudSat. Therefore, we suggest a correction for profiles below 1.5 km that show fractions above 30 % which is simply a function of cloud fraction. In this way, the overestimation can be corrected roughly with a residual uncertainty of 5 pp. Note that cloud fractions and CloudSat's performance might depend on flight tracks.

CloudSat's blind zone underestimates the total precipitation amount by 51 pp, which is mainly light precipitation during CAOs. Moreover, CloudSat's pulse length merges layers of multilayer clouds, thus the amount of multilayer clouds obtained by MiRAC (48 %) reduces by a factor of 4 during the forward simulations.

Additionally, some interesting insights on Arctic low-level clouds have been revealed: Clouds over sea ice showed a rather constant vertical profile while low-level cloud formation strongly enhances cloud fraction over water up to around 1 km. The

cloud fractions obtained by MiRAC indicate that low-level stratus appears at their lowest heights over sea ice. This stratus was also frequently found below 150 m during a Polarstern cruise taking place in parallel to ACLOUD (Griesche et al., 2020). These surface coupled clouds have a strong radiative effect but their spatial extent is mainly unknown due the gaps in the current observation system. Hence, further measurements are needed to study them in more detail (Griesche et al., 2021).

To generalize our findings to the broader Arctic region, further air- or shipborne measurements such as MOSAiC campaign

data, which cover a larger area, have to be studied. Moreover, winter and summer time observations are needed to determine cloud occurrence year-round. To mimic CloudSat observations more accurately, the resolution adaption of the fine resolved radar measurements should comprise an across-track convolution in the future as well. Follow-on studies could test the performance of the EarthCARE CPR to detect Arctic low-level clouds and complement the study of Lamer et al. (2020) for warm marine boundary layer clouds. Compared to the CloudSat CPR, the EarthCARE CPR is more sensitive and has the same range

resolution (500 m; Burns et al., 2016). Due to the higher sensitivity but remaining cloud stretching effect of about 250 m, we expect that it will observe more clouds than CloudSat.

*Data availability.* The MiRAC, AMALi, and AMSR2 ARTIST Sea Ice (ASI) sea ice concentration data (version 5.4), which is provided by the University of Bremen, are accessed via the ac3airborne intake catalog (Mech et al., 2022). The MiRAC measurements during ACLOUD (Mech et al., 2022), AFLUX (Mech et al., 2022), and MOSAiC-ACA (Mech et al., 2022), the cloud top heights from AMALi during

ACLOUD (Kulla et al., 2021a) and AFLUX (Kulla et al., 2021b), and the AMSR2 ASI observations (Melsheimer and Spreen, 2019) are stored on the PANGAEA database. Data that is not yet published is stored on the Nextcloud server of the (AC)[3] project. The marine cold air outbreak indices and circulation weather types are calculated from ERA5 reanalysis data (Hersbach et al., 2020).



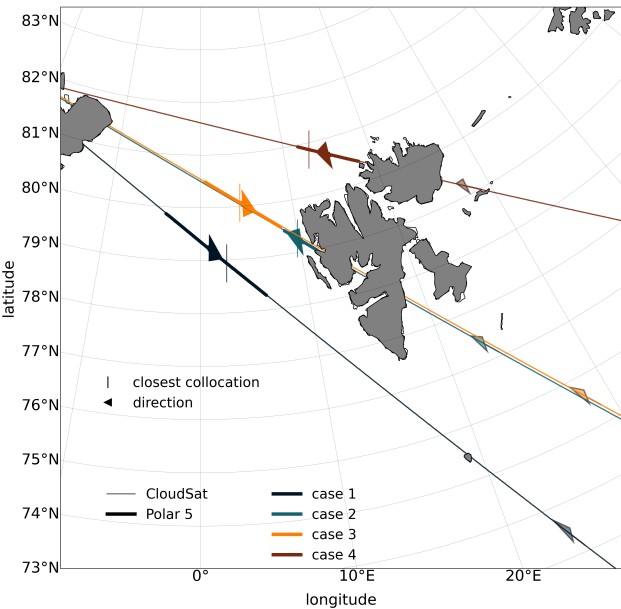

**Figure A1.** Map highlighting the tracks of CloudSat (light colors) and Polar 5 (intense colors) during the four underflights (case 1–4). The arrows and vertical lines indicate the flight direction of each platform and the location of the crossing, respectively.





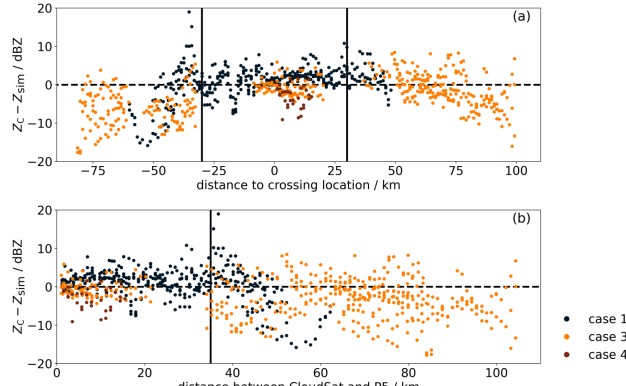

**Figure A2.** Dependence of the difference between the forward-simulated and CloudSat equivalent radar reflectivities ($Z_\mathrm{C} - Z_\mathrm{sim}$) over four underflights of Polar 5 below CloudSat on distance to the crossing location (a) and distance between the platforms (b). Note that CloudSat resolves no signals during case 2.

*Author contributions.* Imke Schirmacher performed the analysis, visualization, writing, and developed and conducted the Methodology. Together with Susanne Crewell and Mario Mech the paper was conceptualized. Pavlos Kollias, Katia Lamer and Lukas Pfitzenmaier provided
the ClouSat expertise and contributed to the algorithm that simulates the CloudSat observations. All authors contributed to manuscript revisions.

*Competing interests.* Pavlos Kollias and Manfred Wendisch are members of the editorial board of Atmospheric Measurement Techniques. The peer-review process was guided by an independent editor, and the authors have also no other competing interests to declare.

*Acknowledgements.* We gratefully acknowledge the funding by the Deutsche Forschungsgemeinschaft (DFG, German Research Foundation)
- Projektnummer 268020496 - TRR 172, within the Transregional Collaborative Research Center "ArctiC Amplification: Climate Relevant Atmospheric and SurfaCe Processes, and Feedback Mechanisms (AC)[3]" - in subproject B03. We thank Sandro Dahlke for providing the marine cold air outbreak indices and Tobias Marke for calculating the circulation weather types for ACLOUD, AFLUX, MOSAiC-(ACA), and HALO-(AC)[3]. We acknowledge the support from the Alfred-Wegener-Institute and Polar 5 captains during the campaigns.





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
