# Peer review of "Assessing Arctic low-level clouds and precipitation from above - a radar perspective"

_EGUsphere, 2023_

## Author Comment (AC1)

**Answers to Reviewer 1**

We thank the reviewer for spending the time and effort to review our study. The comments are very constructive and helped to further improve the manuscript. In this document we reply to every reviewer's comment. The comments of the reviewer are marked in black and our replies in blue. In the revised document, all changes are marked in blue.

1. Line 93:  Are the SIC thresholds chosen arbitrarily, or based on previous studies?  Relatedly, how are these percentages calculated (I assume daily AMSR2 SIC products)?

Previous studies widely used 15 and 80 % to define the marginal ice zone (Strong and Rigor, 2013) following the definition of the World Meteorological Organization (WMO, 1985) for "close ice" (sic > 80 %). In this study, we define sea ice by setting the upper threshold even higher to avoid cloud formation often associated with leads.

We added further information in line 93:

„By using the daily sea ice concentration dataset (version 5.4) obtained by the second Advanced Microwave Scanning Radiometer (AMSR2), we differentiate between open water (sea ice concentration (sic) < 15 %) and sea ice (sic > 90 %). This assumption is more strict than in previous studies (80 %; Strong and Rigor, 2013) to avoid cloud formation associated with leads."

WMO (1985), World Meteorological Organization sea-ice nomenclature, terminology, codes and illustrated glossary, WMO/DMM/BMO 259-TP145. Secretariat of the World Meteorological Organization.

Strong, C. and Rigor, I. G.: Arctic marginal ice zone trending wider in summer and narrower in winter, Geophysical Research Letters, 40, 4864–4868, https://doi.org/10.1002/grl.50928, _eprint: https://onlinelibrary.wiley.com/doi/pdf/10.1002/grl.50928, 2013.

2. Lines 414-416: While I think it is totally acceptable to apply the Maahn et al. (2014) Z-S relation to the MiRAC observations for back-of-the-envelope calcuations, it is likely that the rosette habit assumption used to derive Z-S does not translate very well to the microphysical composition of oceanic snow-producing clouds generated under CAO conditions, especially when comparing snow event categories differentiated by snowrate intensity. Acknowledging this methodological shortcoming is advised, but its overall effect does not detract from the larger message conveyed in the manuscript.

We absolutely agree with the reviewer and added a note in line 415.

„Note that rosette habits might not capture the microphysical composition of oceanic snow-producing clouds under CAO conditions very well."

3. Line 432: ClaudSat → CloudSat

We corrected the typo.

4. Line 470: Kulie et al. (2016) and Kulie and Milani (2018) partition CloudSat-observed snow events by "shallow" and "deep" categories, with special emphasis on high latitude regions prone to CAO's. They highlight the light nature of shallow snow in CAO regions with appropriate (but unresolved) blind zone related caveats. This study clearly indicates that CloudSat estimated

snowfall occurrence and rate/amount are significantly impacted by blind zone limitations that hamper efforts to quantify snowfall with the best available spaceborne instruments.

This is a nice additional information for the manuscript. We modified line 470 as follows:

„This study confirms the finding of Kulie et al. (2016) and Kulie and Milani (2018) that CloudSat observes mainly light snow events in high latitudes during CAOs. The previous studies highlight the by then unresolved blind zone limitations. This study resolves the caveats on snowfall occurrence and amount that lead to an underestimation of the total precipitation amount by 51 pp. This finding hampers efforts to quantify snowfall, especially light one during CAOs, with the best available spaceborne instruments."

5. General comment: It might be worth mentioning that a combined CloudSat/CALIPSO product exists that will more successfully identify low-level cloud structures compared to the CloudSat 2B-Geoprof product.

The reviewer is right. We added a comment on the DARDAR product and explain why we do not use this product in line 125.

„Contrary to this study, Mioche et al. (2015) investigate the radar-lidar combined product DARDAR that might more successfully identify low-level cloud structures compared to the '2B-Geoprof' product. However, DARDAR interpolates the CPR data in the vertical to the finer resolution of the lidar (Winker et al., 2003), still detects ground clutter erroneously as near-surface supercooled droplets, and thus overestimates surface near cloud fraction (Blanchard et al., 2014)."

6. General comment: Mateling et al. (2023; JGR) was just published. It focuses on CAO snowfall production in the North Atlantic Ocean using CloudSat products - another highly relevant manuscript that would benefit from the information gained from the current study.

Thanks a lot for this comment. We will get in touch with the corresponding authors and discuss the implications on their results.

---

## Author Comment (AC2)

**Answers to Reviewer 2**

We thank the reviewer for spending the time and effort to review our study. The comments are very constructive and helped to further improve the manuscript. In this document we reply to every reviewer's comment. The comments of the reviewer are marked in black and our replies in blue. In the revised document, all changes are marked in blue.

Line 142-143: Please rephrase the end of the sentence

We rephrase line 142-143 as follows:

„For this study, we accessed all airborne data via the ac3airborne module that, among other things, stores all links to the data (Mech et al., 2022)."

Line 432 : replace "ClaudSat" by "CloudSat"

We corrected the typo.

Figure 3: You could add the histogram with all data (separated by CWT)?

We added a fourth group of histograms to Figure 3 that consists of all data separated into the four CWT classes and an additional remark in the caption.

„Analyzed flight hours during the different Circulation Weather Types (CWTs) for each campaign and over all campaigns."

Figure 7 and lines 288-289: I suppose the 2 dB difference you mention between $Z_{sim}$ and $Z_C$ comes from a linear fitting ? I suggest you add it on the figure.

Thanks a lot for this suggestion, you are totally right. We added the linear fit to Figure 7 and an additional remark in the caption.

„Comparison of the equivalent radar reflectivity obtained from forward simulations $Z_{sim}$ and CloudSat $Z_C$ over four underflights of Polar 5 below CloudSat with the corresponding linear fit (black). The bin size equals 2 dBZ."

Line 410: what is the assumption of crystal habit for Z-S law in the 2C-Snow-Profile product? Please add a few details on this product to facilitate de further discussion on the differences.

We appreciate the comment of the Referee. The snowfall rate of the 2C-Snow-Profile product is not directly calculated with help of a specific $Z_e$-S relationship. Instead, snow size distribution parameters and their uncertainties are retrieved for radar bins that contain snow and snow-producing clouds via optimal estimation from which the snowfall rate is derived. Wood (2011) confirms that the results represents a range of scene-dependent $Z_e$-S relations. Radar reflectivity profiles of the '2B-Geoprof' product, temperatures from ECMWF-AUX, and a priori snow microphysical properties, radar scattering properties, and size distribution parameters serve as input (Wood et al., 2018). A priori microphysical parameters, scattering properties and their uncertainties are obtained from field campaign measurements that represent dry snow (Wood, 2011). Radar backscattering and extinction cross-sections are calculated with a dipole model for irregularly-shaped particles. The snowfall rate of the 2C-Snow-Profile product is then calculated

from these retrieved snow size distribution parameters and uncertainties via optimal estimation as well. If the surface precipitation type is snow, the surface snowfall rate is estimated using the snow properties at the base of the snow layer which is at least inside the near-surface bin but not closer to ground due to ground clutter.

We added the following in line 411: "The snowfall rate of the 2C-Snow-Profile product is calculated for bins that contain snow or snow-producing clouds via optimal estimation from snow size distribution parameters and uncertainties that are obtained by optimal estimation as well (Wood and L'Ecuyer, 2018). To calculate these snow size distribution parameters, radar reflectivity profiles of the '2B-Geoprof' product, temperatures from ECMWF-AUX and a priori snow microphysical properties, radar scattering properties, and size distribution parameters are required as input. These microphysical parameters represent dry snow and the scattering properties hold for irregularly-shaped particles (Wood, 2011)."

Wood, N. B., 2011: Estimation of snow microphysical properties with application to millimeter-wavelength radar retrievals for snowfall rate. Ph.D. dissertation, Colorado State University, 248 pp. [Available from Colorado State University, Digital Collections, http://hdl.handle.net/10217/48170].

Wood, N. B. and T. S. L'Ecuyer, 2018: Level 2C Snow Profile Process Description and Interface Control Document, Product Version P1 R05. NASA JPL CloudSat project document revision 0., 26 pp. Available from [https://www.cloudsat.cira.colostate.edu/cloudsat-static/info/dl/2c-snow-profile/2C-SNOW-PROFILE_PDICD.P1_R05.rev0_.pdf].

Figure 12: I suggest adding the profile of $S_M$ along that of $A_{M,norm}$.

We added the $S_M$ profile to Figure 12 and the following sentence to the Figure caption:

„The profile of $S_M$ with height is shown in the right column."

Moreover, we added a reference to Fig. 12 in line 428: „Over all campaigns, the total precipitation amount obtained from MiRAC ($A_M$) is 1.0 mm ($S_M$ of 111 mm yr$^{-1}$) at 1.2 km and with 2.1 mm ($S_M$ of 229 mm yr$^{-1}$) more than twice as much at 150 m (Fig. 12)."